# Emergency Department Mean Physician Time per Patient and Workload Predictors ED-MPTPP

**DOI:** 10.3390/jcm9113725

**Published:** 2020-11-20

**Authors:** Julian Wrede, Helge Wrede, Wilhelm Behringer

**Affiliations:** 1Department of Emergency Medicine, Faculty of Medicine, University of Jena, Am Klinikum 1, 07747 Jena, Germany; julianwrede@web.de; 2Höltystraße 9, 22085 Hamburg, Germany; helge.wrede@gmail.com

**Keywords:** manpower, emergency department, physician workload, staffing, treatment

## Abstract

One key element for emergency department (ED) staff calculation is the mean physician time per patient (MPTPP) and its influencing factors. The aims of this study were measuring the MPTPP, identifying factors with significant influence on the MPTPP, and developing a model to predict the MPTPP. This study was a prospective trial conducted at the ED of a university hospital in Germany. The MPTPP was measured with a specifically developed app. The influence of different factors on MPTPP were first tested in univariate analysis. Then, all significant factors were used in a multivariant regression model to minimize collinearities and to develop a prediction model. In total, 202 patients treated by 32 different physicians were observed within one year. The MPTPP was 47 min (standard deviation: 34 min). Relevant factors influencing the MPTPP were treatment area, Emergency Severity Index (ESI) triage level, guiding symptom category, and physician level (all *p* < 0.001). This model predicted 45% of the variance in the MPTPP (*p* < 0.001), which corresponds to a large effect size. We developed an effective prediction model for ED MPTPP, resulting in an MPTPP of 47 min. Future studies are needed to validate our model, which could serve as a benchmark for other EDs where the MPTPP is not available.

## 1. Introduction

Crowding in emergency departments (ED) increases the risk of preventable medical errors [1], increases mortality rate [2], and decreases health care quality [3]. The calculation of the optimal number of staff required might be one possible important factor to ensure patient flow and to avoid crowding.

Different models for the calculation of the optimal number of staff required can be used [4]. For most of these calculation models, it is crucial to know the mean physician time needed per patient (MPTPP). Surprisingly, only a few studies on this important element of staff calculation have been published. Real-time measurement of the MPTPP was provided in two studies in Canada [5,6] and in one study in the USA [7], reporting an MPTPP between 19 and 25 min for fully trained specialist physicians (doctors in training were not included in the study). In Germany, one study estimated the MPTPP by applying benchmarks from anesthesia treatment, calculating an MPTPP of 30 min [8]. Another study evaluated the MPTPP by physician interviews, reporting an estimated MPTPP of 33 min for fully trained specialist physicians and 44 min for doctors in training in Germany, and 38 min for fully trained specialist physicians and 50 min for doctors in training in France [9].

To our knowledge, there is no European study which actually measured MPTPP in the emergency department. Therefore, the aims of this study were measuring the MPTPP, identifying factors with a significant influence on the MPTPP, and developing a model to predict the MPTPP.

## 2. Experimental Section

### 2.1. Design and Setting

This prospective cohort study was conducted in the ED of a university hospital in Germany. The ED sees approximately 33,000 patients per year and consists of three resuscitation rooms, 14 treatment spaces with stretchers and monitoring, one procedure room, one room for casting, and eight rooms for walking patients (fast-track area), including one room with special ears-nose-throat (ENT) equipment, and one room with special ophthalmic equipment. In total, the ED employs 11 fully trained specialist physicians and 14 doctors in training. Specialty emergency medicine does not exist in Germany; thus, fully trained specialist physicians come from various backgrounds, such as internal medicine, surgery, anesthesiology, or cardiology, as well as intensive care medicine. Since the fully trained specialist physicians have worked for many years in the ED, they have gained vast knowledge in emergency medicine and have seen patients with all kinds of symptoms. The doctors in training are either in their family medicine residency and stay for 18 months in the ED, or in their internal medicine or surgery, or trauma surgery residency and stay for 6 months in the ED. All patients are seen by doctors in training regardless of the patients’ symptoms or the specialty of the doctors in training. The doctors in training are all supervised by fully trained specialist physicians of the ED. In cases of higher patient volume, fully trained specialist physicians see patients without doctors in training to speed up the process.

### 2.2. Study Procedures

The MPTPP was measured with a smartphone application designed especially for this study. It allowed for the simultaneous allocation of different physicians to multiple patients, which in turn allowed for the documentation of the time required by each physician to work with one patient. Whenever a physician performed a patient-related activity, the time was started and stopped accordingly. After the case was closed and no more patient-related activity was performed, the total time spent per patient was recorded as total physician time for that specific patient. The MPTPP is the sum of the treatment times each emergency physician took for one particular patient.

The treatment time consists of the time needed for the following actions: taking patient history; examination; charting and documentation; ordering tests and treatments; communication with the patient, family members, or other health professionals; reviewing interviews and test results; carrying out procedures; providing direct bedside care; and looking up information and administrative functions related directly to the patient, such as searching for available beds for admission. The time needed by a consulting physician from another specialty was not measured and not included in the calculations. The MPTPP did not include the waiting time of the patient.

Fully trained specialist physicians and doctors in training were included in the study. The assignment of physicians to patients was random. If a certain patient was treated by a doctor in training and a fully trained specialist physician working in the capacity of a treating physician, the total MPTPP was assigned to the physician, who spent more time with the patient. The MPTPP does not include time for supervising a doctor in training.

### 2.3. Sample Size

The number of patient observations needed was calculated by the rule of ten, as suggested by Everitt [10]. The distribution of observations regarding treatment area, time of the day, and weekday was matched with the overall distribution of the patient volume in the ED.

### 2.4. Influencing Factors

Prior to the study, a set of potentially relevant influencing factors was determined, based on the factors already mentioned in the literature and clinical experience. These factors included the following: the patient’s age in years; the patient’s gender; whether the patient was treated in the fast-track area or acute area; Emergency Severity Index (ESI) triage level by arrival; time of the day, with the beginning of the treatment at 00:00–07:59, 08:00–15:59 or 16:00–23:59, guiding symptom category, with guiding symptom as described in the Canadian Emergency Department Information System (CEDIS) Presenting Complaint List [11] and as categorized in internal, traumatological, neurological, and other guiding symptoms; day of the week; physician level (whether the doctor was a doctor in training or a fully trained specialist physician); and need of a consultation of a specialty not present in the ED.

### 2.5. Ethical Considerations

The study involved no experimental treatments and posed no risk to patients, so the university ethics committee approved the study without the need for patient consent. All emergency physicians gave verbal consent for observation, time measurement, and data collection.

### 2.6. Statistical Analysis

Data was collected by the smartphone application and entered into an Excel spreadsheet. Statistical analyses were done using SPSS Statistics (V. 24.0.0.1, IBM, Armonk, NY, USA).

For analysis, a descriptive statistic and a significance test were carried out. Significance level was determined as *p* < 0.05. Depending on the parameters, different tests were performed:Mann-Whitney test for two independent, non-parametric samples;Linear regression for interval scaled variables;Kolmogorov-Smirnov Goodness-of-Fit test for the distribution analysis of an interval scaled variable;Pearson’s chi-squared test for two categorical variables to test if the frequency distribution of a non-dichotomous variable corresponds to an expected distribution;Kruskal-Wallis test for more than two independent, non-parametric samples;Dunn-Bonferroni tests as post-hoc tests after a significant Kruskal-Wallis test;Binominal test to assess if the frequency distribution of a dichotomous variable corresponds to an expected distribution;A generalized linear model to create the multivariant model used to estimate the MPTPP of a patient and to identify collinearities between significant factors.

First, all potentially relevant influencing factors were analyzed using a univariate analysis. In the second step, significant factors were used to create a model. Non-significant factors were excluded one after another until only significant factors remained in the model. Subsequently, the Bayesian information criterion (BIC) was used to identify factors which could be excluded to improve the goodness of fit. The lower the BIC, the better the goodness of fit [12].

The coefficient of determination for the multivariant model is *R*^2^. It is the proportion of the variance in the dependent variable that can be predicted from the independent variables [13]. With *R*^2^, the effect size of the model can be calculated. Therefore, Cohen’s $f^{2}$ can be calculated with the formula $f^{2}=\frac{R^{2}}{1-R^{2}}$. Here, 0.02 is equivalent to a small, 0.15 to a medium, and 0.35 to a large effect size [13].

## 3. Results

In total, 32 physicians were included in the study (11 fully trained specialist physicians and 21 doctors in training). A total of 142 patients were seen by doctors in training, 49 patients were seen by fully trained specialist physicians, and 11 patients were seen by doctors in training and fully trained specialist physicians. In 8 of these 11 patients, doctors in training spent the majority of the time with the patients. Therefore, the total MPTPP of these 8 patients was assigned to the doctors in training. Thus, in the analysis, a total of 150 patients were seen only by doctors in training, and 52 patients were seen only by fully trained specialist physicians.

Table 1 summarizes the baseline characteristics of the patients. Only two observed patients arrived with ESI level 1. Thus, level 1 and 2 had to be combined to reach an adequate number of patients. These characteristics reflected the department’s overall patient mix, suggesting that a representative sample was included.

The overall MPTPP was 47 min (SD: 34 min).

The following factors showed statistically significant differences in MPTPP (Table 1): treatment area, ESI triage level, arrival time, guiding symptom category, day of the week, physician level, possible consultation, and age. The patient’s gender had no significant influence on MPTPP.

In the multivariant regression model in the first step, the following factors were excluded due to a lack of significance: patient age, arrival time, and necessity of consultation. This lack of significance is mostly caused by collinearities between the factors. In the second step, the factor day of the week was excluded, reducing the BIC (from 1931 to 1920) and improving the goodness of fit. The following regression equation was derived from these data:$$\mathrm{MPTPP}\;=\;25\;+\;13\;\times \;X_{1}\;-\;27\;\times \;X_{2}\;+\;27\;\times \;X_{3}\;+\;15\;\times \;X_{4}\;+\;6\;\times \;X_{5}\;+\;22\;\times \;X_{6}+\;13\;\times \;X_{7}\;+\;10\;\times \;X_{8}$$ where $X_{1}$ describes the physician level, $X_{2}$ describes the treatment area, $X_{3}-X_{5}$ describe the ESI triage level, and $X_{6}-X_{8}$ describe the guiding symptom category.

Table 2 shows in detail which variable has to be replaced with which number.

## 4. Discussion

Our study found an MPTPP of 47 min for ED patients, with treatment area, ESI triage level, guiding symptom category, and physician level being the most relevant factors for the MPTPP. We developed a prediction model which can calculate the MPTPP for individual EDs, taking into account the varying influence of each individual factor.

To our knowledge, this is the first real-time measurement of the MPTPP for ED patients in Europe. Medical training and care delivery are not homogenous in Europe. Every country has differences in how much supervision doctors in training receive and the amount of autonomy they are allowed. Therefore, we did not include time for supervision or teaching in the MPTPP. First, the MPTPP for patient care has to be determined to be able to calculate the optimal number of staff required for safe patient care. In the next step, time for supervision and training needs to be determined, which will depend on the setting of each individual emergency department. Whereas the measured MPTPP of 52 min for doctors in training in our study reflects the MPTPP for doctors in training found in interviews in Germany and France [9], the measured MPTPP of 31 min for fully trained specialist physicians in our study is shorter compared to the MPTPP for fully trained specialist physicians found in interviews in Germany and in France [9], and longer compared to the measured MPTPP for fully trained specialist physicians in the studies from Canada and the USA [5,6,7]. Interviews have the limitation of subjective estimations, which could be one explanation for the considerably longer MPTPP in this interview-based study [9]. In the USA and Canada, emergency medicine is a well-established specialty with a formally structured residency program, whereas in Germany, emergency medicine is not a specialty. Thus, no formal structured training in emergency medicine exists in Germany. It is debatable if this difference in emergency-medicine training is causing the longer MPTPP measured in our study. Moreover, since many factors influence the MPTPP, as shown in our study and others [5], comparisons between single EDs need careful interpretations. Unmeasurable factors like documentation systems or communication culture also need to be taken into account when comparing the MPTPP. Future studies should measure and analyze the various components of the MPTPP. Tasks such as trying to get hold of colleagues or obtaining information and making sure it is correct should be analyzed in more detail to identify potentials for more efficient patient care. Although the factor of physician level influenced the MPTPP in our study, the impact of this factor on the total MPTPP was less than that of other factors and lower than anticipated. However, the difference in MPTPP between fully trained specialist physicians and doctors in training was 21 min.

In our study, the factor treatment area was one of the two most relevant factors for the MPTPP. Patients in the fast-track showed a 40 min shorter MPTPP than patients in the acute area. This difference can be explained by the lower complexity and hence by the lower time investment in these patients. Though the MPTPP was shorter in the fast-track area as compared to the acute area, it has to be recognized that the occasional alert, mobile patient with normal vital signs who is seriously ill often require more clinical skill than the emergency management of those who are obviously sick. Another study found “arrival by ambulance” as strongest predictor for MPTPP [5]. We did not investigate this specific factor, but if we assume that patients arriving in an ambulance are in general sicker and thus put in the acute area, then our results would be in line with this study.

The other most relevant factor for the MPTPP in our study was the factor of ESI triage level 1 and 2. The ESI triage algorithm yields rapid, reproducible, and clinically relevant stratification of patients into five groups, from level 1 (most urgent) to level 5 (least urgent) [14]. With increasing urgent triage level, the MPTPP increased, with a difference of 58 min between patients with the least urgent category 5 and the combined most urgent triage categories 1 and 2. Our results are in line with studies from the USA and Canada, which also found that triage level was associated with workload [5,6].

Guiding symptom category was another factor for the MPTPP in our study, with symptoms from the internal medicine category having the greatest impact. Although other studies did not investigate the guiding symptom category specifically, surrogate parameters for patients with symptoms from the internal medicine category, such as age and comorbidity, were also associated with longer MPTPP [5]. Interestingly, in our study, patient age itself was not a factor for the MPTPP. In the same study mentioned above, procedures were also associated with longer MPTPP [5], whereas in our study, we did not specifically differentiate between ED procedures. However, we found a shorter MPTPP in patients with symptom group traumatology. Either patients with symptoms from simple traumatological procedures such as internal medicine and neurology seem to require more time, or patients with symptom complex traumatology do not always require procedures.

The factors included in our model explain 45% of the variance of the MPTPP (*R*^2^). This corresponds to a Cohen’s f^2^ of 0.82 which is equivalent to a large effect size [13]. The only other study formulating a model for predicting the MPTPP showed a predicted variance of 31% [5]. It seems that the factors we used in our model allow a more precise prediction of workload. Future studies might evaluate a combination of factors. We could not evaluate all of the factors from other studies, because they were not available to us. In environments with a shortage of nursing staff, certain non-physician tasks might be performed by physicians, thus increasing MPTPP. In environments with nurse practitioners as part of the team, the number of patients seen by a physician and/or MPTPP might decrease.

Formulas for staff calculation use number of patients and workload per patient to obtain the needed number of full-time equivalents [15]. From our study, it is obvious that workload per patients and MPTPP can vary considerably, mostly depending on patient mix and staff mix. The most accurate staff calculation would include the actual measurement of MPTPP for each individual ED, but these measurements are time and resource consuming. If the MPTPP were not available for individual EDs, the second-best alternative would be using benchmarks from the literature. Once validated, the results of our study could serve other EDs as a benchmark. The provision of a model rather than a fixed time duration would allow individual EDs to calculate their own MPTPP based on distribution of the patients’ treatment area, triage category, and symptom group as well as ratio of fully trained specialist physicians to doctors in training. These parameters are normally easily available through the hospital information system and staff plan, and could be integrated into our formula. Common calculation tools using queueing theory, incorporating the MPTPP, could then be used for individual calculation of the optimal number of staff required [15]. A calculation of the optimal number of staff required based on objective criteria, such as number of patients and a benchmark MPTPP, might help to convince hospital administrators to increase the budget to match patient volume with an adequate number of staff.

Future studies are needed to investigate the MPTPP in various settings and types of emergency departments to identify more factors impacting the MPTPP. Such factors could include whether the hospital is a university hospital or non-university hospital, city hospital or rural hospital, and referral hospital or general hospital. Moreover, the MPTPP could be compared between states within one country or even between countries. If differences in the MPTPP are detected, factors leading to these differences need to be identified to understand how the MPTPP can be optimized.

## 5. Conclusions

We developed an effective prediction model for ED MPTPP and identified the following relevant factors in order of decreasing effect: treatment area, ESI triage level, guiding symptom category, and physician level, resulting in an MPTPP of 47 min. The complexity of determining the MPTPP as a key element of proper staff calculation needs to be considered. Future multicentric studies are needed to validate our model, which could serve as a benchmark for other EDs where the MPTPP is not available.

## 6. Limitations of the Study

First of all, the study was not blinded. The physicians were aware that they were observed to measure the time used for patient diagnosis and treatment. Therefore, a willingly prolonged duration of allocated time to prove a shortage of workforce, or shorter duration of allocated time to prove efficacy of a single doctor cannot be excluded.

The study was monocentric and our results may not apply to other sites. However, we are confident that we could partly overcome this limitation by including a wide range of physicians with different education levels and experience.

Pediatric patients and patients with gynecological, ophthalmic, dermatologic, psychiatric, and ENT symptoms were not included in the study.

Other factors, such as comorbidity and number of previous visits were not measured. Further factors should be investigated to increase the effectiveness of the study.

It is possible to generate a negative MPTPP in our model. Although our model has a large effect size, some uncertainty concerning the MPTPP remains. Certain constellations of the factors included in the model will thus result in a negative MPTPP. This weakness is shared with the model developed by Innes et al. [5].

Lastly, due to a limited number of observers, it was not possible to generate a higher number of patient observations, which prevented a validation of our results.

## Figures and Tables

**Table 1 jcm-09-03725-t001:** Characteristics of the data sample, including the impact and significance level of the factors influencing the mean physician time per patient (MPTPP) in univariate analysis.

Factor	No. (and %) *	Minutes (SD)	*p*
Age, year; mean (SD)	54 (23)		<0.001
Gender			0.659
Male	112 (55)	48 (37)	
Female	90 (45)	44 (30)	
Treatment area			<0.001
Fast-track	123 (61)	31 (17)	
Acute	79 (39)	71 (39)	
ESI triage level			<0.001
1 + 2	45 (22)	76 (42)	
3	93 (46)	48 (27)	
4	43 (21)	27 (15)	
5	21 (10)	18 (11)	
Arrival time			0.01
08:00–15:59	127 (63)	43 (28)	
16:00–23:59	59 (29)	46 (35)	
00:00–07:59	16 (8)	79 (51)	
Guiding symptom category			<0.000
Internal	74 (37)	57 (41)	
Traumatology	73 (36)	39 (24)	
Neurological	38 (19)	50 (33)	
Others	17 (8)	25 (13)	
Day of the week			0.002
Monday	22 (11)	43 (19)	
Tuesday	42 (21)	34 (19)	
Wednesday	34 (17)	56 (40)	
Thursday	32 (16)	34 (20)	
Friday	27 (13)	62 (33)	
Saturday	25 (12)	52 (47)	
Sunday	20 (10)	53 (43)	
Physician level			<0.001
Doctor in training	150 (74)	52 (36)	
Fully trained specialist physicians	52 (26)	31 (16)	
Consultation needed			<0.001
Yes	52 (26)	59 (35)	
No	150 (74)	42 (32)	

*n* = 202; *: unless otherwise specified; SD: standard deviation; ESI: Emergency Severity Index.

**Table 2 jcm-09-03725-t002:** Replacing variables for the factors of the prediction model.

Factor	Characteristics	Value
Physician level	Doctor in training	$X_{1}\;$= 1
	fully trained specialist physicians	$X_{1}\;$= 0
Treatment area	fast-track	$X_{2}$= 1
	non-fast-track	$X_{2}$= 0
ESI triage level	1 + 2	$X_{3}\;$$=\;1,\;X_{4\;}$$=\;0,\;X_{5}\;$= 0
	3	$X_{3}\;$$=\;0,\;X_{4}\;$$=\;1,\;X_{5}\;$= 0
	4	$X_{3\;}$$=\;0,\;X_{4\;}$$=\;0,\;X_{5}\;$= 1
	5	$X_{3\;}$$=\;0,\;X_{4}\;$$=\;0,\;X_{5}\;$= 0
Guiding symptom category	Internal	$X_{6}\;$$=\;1,\;X_{7}\;$$=\;0,\;X_{8}\;$= 0
	Traumatology	$X_{6}\;$$=\;0,\;X_{7\;}$$=\;1,\;X_{8}\;$= 0
	Neurological	$X_{6}\;$$=\;0,\;X_{7}\;$$=\;0,\;X_{8}\;$= 1
	Others	$X_{6\;}$$=\;0,\;X_{7\;}$$=\;0,\;X_{8\;}$= 0

*R*^2^ of the model was 0.45 (*p* < 0.001).

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
