# Peer review of "Emergency Department Mean Physician Time per Patient and Workload Predictors ED-MPTPP"

_jcm, 2020, doi:10.3390/jcm9113725_

Round 1
Reviewer 1 Report
Great job addressing the comments.
Author Response
Thank you very much for your comment.
Reviewer 2 Report
This is an interesting study that addresses and important clinical issue that needs further study. My main concern is definitions of terms – in particular what is meant by a “consultant” and a “resident”, and if patients were managed by individual doctors, or “teams” of physicians providing care in parallel and contemporaneously, or in shifts with a lot of management time consumed in hand-overs etc.
Overall, the paper was will written, but nevertheless some things are unclear.
Introduction:
- “Adequate staff calculation might be one possible important factor to ensure patient flow and to avoid crowding.” This is unclear. Does this mean “provision of enough staff might improve patient flow and avoid over-crowding”
- What does “staff calculation” mean? Does it mean calculate the optimal number of staff required?
Methods
3.The terms consultant and resident should be defined, as they are meaningless to many. I suggest they are replaced by “fully trained specialist physicians” and “doctors in training”.
It is important to clarify if “residents” have clinical autonomy – i.e. can they manage a patient without approval of a “consultant” OR do residents always need approval for ALL or only some management decisions such as discharge from or admission to hospital. I suspect it depends on the years of training and the assessed competence of the “resident” in question.
I wonder if this study would be wiser to confine itself to physicians who were completely autonomous, or at least to physicians who were competent to make the decision to either discharge patients or admit them to hospital.
4.“real-time in a controlled experimental setting” What does this mean? It was not done in real-life practice, but as a simulation. Surely not, please explain.
5.What are “Positions”? Are they cubicles, or alcoves, or corridors?
6.Were doctors treating only one patient at a time, or were they seeing other patients at the same time? So, a doctor would see patient A and order a test, and then see patient B while wating for patient A’s results. Therefore, did the time measured only include the time spent WITH patient A, and not the time it took for the result to come back. OR was the MPTPP the entire time spent with patient A including wait times. This needs to be made CRYSTAL clear, maybe with a worked through example.
7.“Assignment of physician to patients was random. If a certain patient was seen by a resident and a consultant, total MPTPP was assign to the physician, who spent longer time with the patient.”
So, does this mean that residents and consultants were included in the study? Inclusion criteria for participants should be clear. Also, does this mean that many patients were co-managed by more than one physician? Does this only apply to management by a consultant and resident, or did it also include two or more consultants sharing a patient, or a consultant sharing care with several (i.e. a team of) residents?
Results
8.“202 patients treated by 32 different physicians were analysed.” Does that mean that each physician managed 6.3 patients alone and without assistance from any other physician, or did 32 physicians collectively manage all 202 patients, or is it somewhere between?
9.Table 1 I do not understand p values are they misaligned? How can an age of 54 have a p of <0.0001? Compared to what?
- Statistical question: As I understand it 8 categorical variables have been found to retain their statistical significance for the prediction of the continuous variable MPTPP based on data from only 202 patient encounters. The variables identified make intuitive sense, but I wonder if it is statistically valid. If more patients had been studied would more or fewer variables have been identified by logistic regression? Could the authors comment?
Discussion
11.“To our knowledge this is the first real time measurement of the MPTPP for ED patients in Europe.” Medical training and care delivery are not homogenous in Europe. Every country has differences in how much supervision doctors in training get and the amount of autonomy they are allowed. Indeed, in many jurisdictions the amount of autonomy of a trainee is determined entirely by their supervisor, whereas in others senior decision makers make all, or approve all, substantive management decisions. Clearly the time taken to manage a patient who is being seen by a trainee will take a lot longer of much of that time is taken up reviewing decisions and teaching. All these issues need to be clarified. For example, the statement “…longer as compared to the measured MPTPP for consultants in the studies from Canada and the USA [5-7].” is not surprising, since there are no “consultants” in Canada or the USA, and outside of academic centres most fully trained “specialists” are not involved in training doctors, and are not “assisted” by them. This weakens the argument that they see patients quicker because they are trained in emergency medicine. It could be that they are not slowed down by having to teach and instruct doctors in training.
12.“Also unmeasurable factors like documentation systems or communication culture need to be taken into account, when comparing the MPTPP.” More should be made of this: how much time is spent trying to get hold of colleagues or obtaining information and making sure it is correct?
13.“Patients in the “fast-track” showed a 40 minutes shorter MPTPP than patients in the acute area. This difference can be explained by the lower complexity and hence by the lower time in-vestment in these patients.” Were more or fewer “fast-track” patients managed by doctors in training? This is important, because these low-risk patients often turn out to be high-risk if something rare is missed. Paradoxically recognizing the very occasional alert, mobile patient with normal vital signs who is seriously ill often require more clinical skill than the emergency management of those who are obviously sick.
Author Response
Please see the attachement.
We would like to thank the reviewer for the in-depth review of the manuscript and the helpful remarks and suggestions. Below please find our point by point answers to the reviewer.
This is an interesting study that addresses and important clinical issue that needs further study. My main concern is definitions of terms – in particular what is meant by a “consultant” and a “resident”, and if patients were managed by individual doctors, or “teams” of physicians providing care in parallel and contemporaneously, or in shifts with a lot of management time consumed in hand-overs etc.
Overall, the paper was will written, but nevertheless some things are unclear.
Introduction:
- “Adequate staff calculation might be one possible important factor to ensure patient flow and to avoid crowding.” This is unclear. Does this mean “provision of enough staff might improve patient flow and avoid over-crowding”
- What does “staff calculation” mean? Does it mean calculate the optimal number of staff required?
Reply 1,2: Thank you very much for providing a proposal for clarification. The proposal of the reviewer is exactly, what we wanted to say. We changed accordingly into: “Calculation of the optimal number of staff required and provision of enough staff might improve patient flow and avoid over-crowding”.
Methods
- The terms consultant and resident should be defined, as they are meaningless to many. I suggest they are replaced by “fully trained specialist physicians” and “doctors in training”. It is important to clarify if “residents” have clinical autonomy – i.e. can they manage a patient without approval of a “consultant” OR do residents always need approval for ALL or only some management decisions such as discharge from or admission to hospital. I suspect it depends on the years of training and the assessed competence of the “resident” in question. I wonder if this study would be wiser to confine itself to physicians who were completely autonomous, or at least to physicians who were competent to make the decision to either discharge patients or admit them to hospital.
Reply 3: we changed “consultant” and “resident” accordingly in the entire manuscript as the reviewer suggested. Doctors in training can manage a patient without approval of a supervising specialist physician. Depending on the knowledge of the doctor in training and/or complexity of the patient, the doctors in training will contact the supervising fully trained specialist physician for advice. In our setting, the workload is mainly managed by doctors in training, supervised by fully trained specialist physicians. Thus, we included all doctors in training in the study.
- “real-time in a controlled experimental setting” What does this mean? It was not done in real-life practice, but as a simulation. Surely not, please explain.
Reply 4: We wanted to express that the MPTPP was actually measured for every patient by a student, and not calculated by the number of patients divided by the number of physician-hours (e.g. 100 patients per day and 72 physician hours would result in a calculated MPTPP of 43 minutes). We deleted “real-time in a controlled experimental setting”.
- What are “Positions”? Are they cubicles, or alcoves, or corridors?
Reply 5: we changed “position” to “treatment space”.
- Were doctors treating only one patient at a time, or were they seeing other patients at the same time? So, a doctor would see patient A and order a test, and then see patient B while wating for patient A’s results. Therefore, did the time measured only include the time spent WITH patient A, and not the time it took for the result to come back. OR was the MPTPP the entire time spent with patient A including wait times. This needs to be made CRYSTAL clear, maybe with a worked through example.
Reply 6: The time measured included only the time spent with tasks for the patient, excluding wait times. In the methods section we stated: thus (the app) allowed the documentation of the time required by all physicians working with one patient. Whenever a physician performed a patient related activity, the time was started and stopped accordingly. After the case was closed and no more patient related activity was performed, the total time spent per patient was stored as total physician time for that specific patient. The MPTPP is the sum of the treatment times, each emergency physician took for one particular patient”. We further specified all tasks, which were included in the MPTPP: “The treatment time consists of the time needed for the following actions: taking patient history, examination, charting and documentation, ordering tests and treatments, communication with the patient, family members or other health professionals, reviewing interviews and test results, carrying out procedures, providing direct bedside care, looking up information and administrative functions related directly to the patient such as searching for available beds for admission”. To be more precise, we added following sentence in the methods section: “The MPTPP did not include waiting time of the patient.”
- “Assignment of physician to patients was random. If a certain patient was seen by a resident and a consultant, total MPTPP was assign to the physician, who spent longer time with the patient.” So, does this mean that residents and consultants were included in the study? Inclusion criteria for participants should be clear. Also, does this mean that many patients were co-managed by more than one physician? Does this only apply to management by a consultant and resident, or did it also include two or more consultants sharing a patient, or a consultant sharing care with several (i.e. a team of) residents?
Reply 7: Doctors in training and fully trained specialist physicians were included. Fully trained specialist physicians can function as supervising physician as well as treating physician, as stated in the methods section: “In case of higher patient volume, fully trained specialist physicians see patients without doctors in training to speed up the process.” Eleven patients were co-managed by a doctor in training and a fully trained specialist physician in the function as treating physician. We did not include time of supervision in the MPTPP. “Physician level” was one factor of interest, and in order not to lose these 11 patients for analysis, we assigned the total MPTPP to the physician, who spent longer time with the patient. To be more precise, we changed the paragraph accordingly: “Fully trained specialist physicians and doctors in training were included in the study. Assignment of physician to patients was random. If a certain patient was seen by a doctor in training and a fully trained specialist physician in the function as treating physician, total MPTPP was assigned to the physician, who spent longer time with the patient. The MPTPP does not include time for supervision of a doctor in training.”
Results
- “202 patients treated by 32 different physicians were analysed.” Does that mean that each physician managed 6.3 patients alone and without assistance from any other physician, or did 32 physicians collectively manage all 202 patients, or is it somewhere between?
Reply 8: 32 physicians were included in the study (11 fully trained specialist physicians and 21 doctors in training). 142 patients were seen by doctors in training, 49 patients were seen by fully trained specialist physicians as treating physician, and 11 patients were seen by doctors in training and fully trained specialist physician as treating physician. In 8 of these 11 patients, doctors in training spent the majority of the time with the patients, and therefore the total MPTPP of these 8 patients was assigned to doctors in training Thus, for analysis, a total of 150 patients were seen only by doctors in training, and 52 patients were seen only by fully trained specialist physicians. The results were changed accordingly.
- Table 1 I do not understand p values are they misaligned? How can an age of 54 have a p of <0.0001? Compared to what?
Reply 9: Table 1 shows the results of the baseline variables and the influence on MPTPP in univariate analysis. The mean age was 54 (SD 23) years, and age as continuous variable was statistically significant associated with the MPTPP (p<0.001). Gender was not statistically significant associated with the MPTPP (p=0.659). Treatment area was statistically significant associated with the MPTPP (p<0.001), with patients in the fast track area showing a shorter MPTPP as compared to acute treatment area. We changed the legend for table 1 into: “Characteristics of the data sample, impact and significance level of the factors on the MPTPP in univariate analysis”.
- Statistical question: As I understand it 8 categorical variables have been found to retain their statistical significance for the prediction of the continuous variable MPTPP based on data from only 202 patient encounters. The variables identified make intuitive sense, but I wonder if it is statistically valid. If more patients had been studied would more or fewer variables have been identified by logistic regression? Could the authors comment?
Reply 10: In univariate analysis, 8 variables were statistically significant associated with MPTPP. In the multivariant regression model, the factors “patient age”, “arrival time”, and the “necessity of a consultation” were excluded due to a lack of significance. In general, the power of a study increases with increasing sample size, and as the power increases, there is a decreasing probability of a type II error (wrongly failing to reject the null hypothesis that there is no difference between groups). However, we consulted a statistician, who confirmed that the number of patients was sufficient for the regression model based on reference 10.
Discussion
- “To our knowledge this is the first real time measurement of the MPTPP for ED patients in Europe.” Medical training and care delivery are not homogenous in Europe. Every country has differences in how much supervision doctors in training get and the amount of autonomy they are allowed. Indeed, in many jurisdictions the amount of autonomy of a trainee is determined entirely by their supervisor, whereas in others senior decision makers make all, or approve all, substantive management decisions. Clearly the time taken to manage a patient who is being seen by a trainee will take a lot longer of much of that time is taken up reviewing decisions and teaching. All these issues need to be clarified. For example, the statement “…longer as compared to the measured MPTPP for consultants in the studies from Canada and the USA [5-7].” is not surprising, since there are no “consultants” in Canada or the USA, and outside of academic centres most fully trained “specialists” are not involved in training doctors, and are not “assisted” by them. This weakens the argument that they see patients quicker because they are trained in emergency medicine. It could be that they are not slowed down by having to teach and instruct doctors in training.
Reply 11: we fully agree with the reviewer. Exactly for these reasons we did not include the time for supervision or teaching by senior physicians in the MPTPP. From our perspective, first we have to measure the MPTPP for patient care to be able to calculate the optimal number of staff required for safe patient care. In a next step, time for supervision and training needs to be determined, which of course very much depends on the setting of each emergency department. We added following paragraph in the discussion: “Medical training and care delivery are not homogenous in Europe. Every country has differences in how much supervision doctors in training get and the amount of autonomy they are allowed. Therefore we did not include time for supervision or teaching in the MPTPP. First MPTPP for patient care has to be determined to be able to calculate the optimal number of staff required for safe patient care. In a next step, time for supervision and training needs to be determined, which will depend on the setting of each individual emergency department.”
- “Also unmeasurable factors like documentation systems or communication culture need to be taken into account, when comparing the MPTPP.” More should be made of this: how much time is spent trying to get hold of colleagues or obtaining information and making sure it is correct?
Reply 12: We agree with the reviewer that further studies are needed to measure the different components of the MPTPP to identify potentials for improvement, However, the aim of our study was to measure total MPTPP without measuring each component of the MPTPP. Such a study with detailed measurements would need more resources. We added following paragraph in the discussion: “Future studies, should measure and analyse the various components of the MPTPP. Tasks such as trying to get hold of colleagues or obtaining information and making sure it is correct should be analysed in more detail to identify potentials for more efficient patient care.”
- “Patients in the “fast-track” showed a 40 minutes shorter MPTPP than patients in the acute area. This difference can be explained by the lower complexity and hence by the lower time in-vestment in these patients.” Were more or fewer “fast-track” patients managed by doctors in training? This is important, because these low-risk patients often turn out to be high-risk if something rare is missed. Paradoxically recognizing the very occasional alert, mobile patient with normal vital signs who is seriously ill often require more clinical skill than the emergency management of those who are obviously sick.
Reply 13: We totally agree with the reviewer that the occasional not so obvious sick patient in the fast track area requires more clinical skills than the obvious sick. In the ideal world with unlimited resources, only fully trained specialist physicians should work in the fast track area. However, in general, MPTPP seems to be considerably shorter in the fast track area as compared to the acute area, despite the occasional mobile, but sick patient. According to our results, multivariate analysis showed that the factor “fast track” is independently associated with the MPTPP. To answer the concrete question of the reviewer, out of 123 patients in the fast track area, 84 (68%) patients were managed by doctors in training, and out of 79 patients in the in the acute area, 58 (73%) patients were managed by doctors in training. We added following sentence in the discussion: “Though the MPTPP was shorter in the fast track area as compared to the acute area, it has to be recognized that the occasional alert, mobile patient with normal vital signs who is seriously ill often require more clinical skill than the emergency management of those who are obviously sick.”

Round 2
Reviewer 2 Report
Thank you for your excellent revisions. You have addressed all my concerns
This manuscript is a resubmission of an earlier submission. The following is a list of the peer review reports and author responses from that submission.
Round 1
Reviewer 1 Report
This study examines an important issue -- gathering appropriate data to improve ED staffing, efficiency and effectiveness.
Several comments are offered to improve this manuscript.
- How does mean time for physicians fit into overall ED staffing and patient flow. Is it the only factor are there others- for example, ED nurses. And if there are ED nurses present could their time be seen as a substitute for physician time.
- Why is the sample size so small: 191 patients treated by 32 different physicians were analyzed.
- The study uses what appears to be an efficient tool to gather time data:
The MPTPP was measured with a smartphone application especially designed for this study. It allowed allocating different physicians to multiple patients simultaneously, and thus allowed the documentation of the time required by all physicians working with one patient. The MPTPP is the sum of the treatment times, each emergency physician took for one particular patient.
It seems like this would allow a much bigger sample.
-
Related to sample size. Given the small sample do you expect to have many significant factors in the multivariate model and also given the importance of Fast track/acute and treatment levels would it not make sense to analyze these sub-groups separately.
- Finally, it would be good to add in the discussion how the information in this study is useful and what else would be needed for future research to either fill in the blanks and/or extend the research.
Reviewer 2 Report
In the Methods section, the study setting should be described in more detail. How many beds are in the ED, how many patients are seen annually, and what is the staffing model like? What determines whether a patient sees a Consultant vs. a Resident? More details of the App would also be information; how exactly does the App track time?
Treatment area and ESI category would appear to be collinear with me; patients with ESI 1 and 2 would most likely be seen in the non-fast track treatment area.
Patient comorbidities may definitely influence the outcome; why is this information not available?
In the Discussion, it is unclear how this model would be used in clinical practice. Most EDs are understaffed and overcrowded and the solution would be to increase staff, but this is challenged by budget constraints. How will this model help?